# On the mechanistic nature of epistasis in a canonical *cis*-regulatory element

**Mato Lagator[1†], Tiago Paixão[1†], Nicholas H Barton[1], Jonathan P Bollback[1,2], Călin C Guet[1]***

[1]Institute of Science and Technology Austria, Klosterneuburg, Austria; [2]Department of Integrative Biology, University of Liverpool, Liverpool, United Kingdom

**Abstract** Understanding the relation between genotype and phenotype remains a major challenge. The difficulty of predicting individual mutation effects, and particularly the interactions between them, has prevented the development of a comprehensive theory that links genotypic changes to their phenotypic effects. We show that a general thermodynamic framework for gene regulation, based on a biophysical understanding of protein-DNA binding, accurately predicts the sign of epistasis in a canonical *cis*-regulatory element consisting of overlapping RNA polymerase and repressor binding sites. Sign and magnitude of individual mutation effects are sufficient to predict the sign of epistasis and its environmental dependence. Thus, the thermodynamic model offers the correct null prediction for epistasis between mutations across DNA-binding sites. Our results indicate that a predictive theory for the effects of *cis*-regulatory mutations is possible from first principles, as long as the essential molecular mechanisms and the constraints these impose on a biological system are accounted for.

**\*For correspondence:** calin@ist.ac.at

[†]These authors contributed equally to this work

**Competing interests:** The authors declare that no competing interests exist.

## Introduction

The interaction between individual mutations – epistasis – determines how a genotype maps onto a phenotype (*Wolf et al., 2000*; *Phillips, 2008*; *Breen et al., 2012*). As such, it determines the structure of the fitness landscape (*de Visser and Krug, 2014*) and plays a crucial role in defining adaptive pathways and evolutionary outcomes of complex genetic systems (*Sackton and Hartl, 2016*). For example, epistasis influences the repeatability of evolution (*Weinreich et al., 2006*; *Woods et al., 2011*; *Szendro et al., 2013*), the benefits of sexual reproduction (*Kondrashov, 1988*), and species divergence (*Orr and Turelli, 2001*; *Dettman et al., 2007*). Studies of epistasis have been limited to empirical statistical descriptions, and mostly focused on interactions between individual mutations in structural proteins and enzymes (*Phillips, 2008*; *Starr and Thornton, 2016*). While identifying a wide range of possible interactions (*Figure 1*), these studies have not led to a consensus on whether there is a systematic bias on the sign of epistasis (*Lalić and Elena, 2013*; *Kussell, 2013*; *Velenich and Gore, 2013*; *Kondrashov and Kondrashov, 2015*), a critical feature determining the ruggedness of the fitness landscape (*Poelwijk et al., 2011*). Specifically, it is only when mutations are in sign epistasis that the fitness landscape can have multiple fitness peaks - a feature that determines the number of evolutionary paths that are accessible to Darwinian adaptation (*de Visser and Krug, 2014*). Furthermore, even a pattern of positive or negative epistasis has consequences for important evolutionary questions such as the maintenance of genetic diversity (*Charlesworth et al., 1995*) and the evolution of sex (*Kondrashov, 1988*; *Otto and Lenormand, 2002*). While the absence of such a bias does not reduce the effect of epistasis on the response to selection, it does demonstrate that predicting epistasis remains elusive.

Scarcity of predictive models of epistasis comes as no surprise, given that most experimental studies focused on proteins. The inability to predict structure from sequence, due to the

**eLife digest** Mutations are changes to DNA that provide the raw material upon which evolution can act. Therefore, to understand evolution, we need to know the effects of mutations, and how those mutations interact with each other (a phenomenon referred to as epistasis). So far, few mathematical models allow scientists to predict the effects of mutations, and even fewer are able to predict epistasis.

Biological systems are complex and consist of many proteins and other molecules. Genes are the sections of DNA that provide the instructions needed to produce these molecules, and some genes encode proteins that can bind to DNA to control whether other genes are switched on or off. Lagator, Paixão et al. have now used mathematical models and experiments to understand how the environment inside the cells of a bacterium known as *E. coli*, specifically the amount of particular proteins, affects epistasis.

These mathematical models are able to predict interactions between mutations in the most abundant class of DNA-binding sites in proteins. This approach found that the nature of the interaction between mutations can be explained through biophysical laws, combined with the basic knowledge of the logic of how genes regulate each other's activities. Furthermore, the models allow Lagator, Paixão et al. to predict interactions between mutations in several different environments, such as the presence of a new food source or a toxin, defined by the amounts of relevant DNA-binding proteins in cells.

By providing new ways of understanding how genes are regulated in bacteria, and how gene regulation is affected by mutations, these findings contribute to our understanding of how organisms evolve. In addition, this work may help us to build artificial networks of genes that interact with each other to produce a desired response, such as more efficient production of fuel from ethanol or the break down of hazardous chemicals.

prohibitively large sequence space that would need to be experimentally explored in order to understand even just the effects of point mutations (*Maerkl and Quake, 2009*; *Shultzaberger et al., 2012*), let alone the interactions between them, prevents the development of a predictive theory of epistasis (*Lehner, 2013*; *de Visser and Krug, 2014*). In fact, the only predictive models of epistasis focus on tractable systems where it is possible to connect the effects of mutations to the underlying biophysical and molecular mechanisms of the molecular machinery (*Dean and Thornton, 2007*; *Lehner, 2011*); namely, RNA sequence-to-shape models (*Schuster, 2006*), and models of metabolic networks (*Szathmáry, 1993*). Even though these studies have provided accurate predictions of interactions between mutations, applying their findings to address broader evolutionary questions remains challenging. For RNA sequence-to-shape models, the function of a novel phenotype (new folding structure) is impossible to determine without experiments. In addition, this approach cannot account for the dependence of epistatic interactions on even simple variations in cellular environments, which are known to affect epistasis (*Flynn et al., 2013*; *Caudle et al., 2014*). On the other hand, metabolic network models are limited to examining the effects of large effect mutations, like deletions and knockouts, and lack an explicit reference to genotype.

In order to overcome the limitations of existing theoretical approaches to predicting epistasis, we focused on bacterial regulation of gene expression as one of the simplest model systems in which the molecular biology and biophysics of the interacting components are well understood. We analyze the effects of mutations in a prokaryotic *cis*-regulatory element (CRE) – the region upstream of a gene containing DNA-binding sites for RNA polymerase (RNAP) and transcription factors (TFs). As such, we study a molecular system where an interaction between multiple components, rather than a single protein, determines the phenotype. Promoters that are regulated by competitive exclusion of RNAP by a repressor are particularly good candidates for developing a systematic approach to understanding epistasis as, in contrast to coding regions as well as more complex CREs and activatable promoters (*Garcia et al., 2012*), the phenotypic effects of mutations in binding sites of RNAP and repressor are tractable due to their short length and the well-understood biophysical properties of protein-DNA interactions (*Bintu et al., 2005b*; *Saiz and Vilar, 2008*; *Vilar, 2010*). Understanding

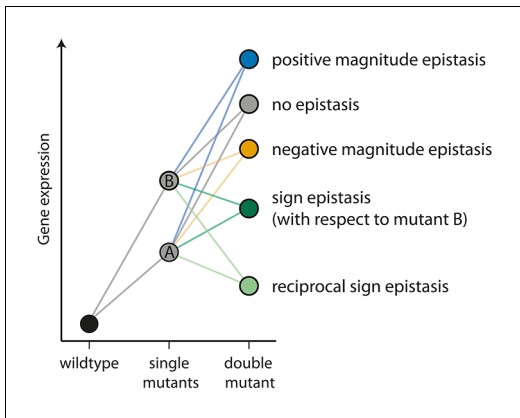

**Figure 1.** The different types of epistasis between two point mutations. Two point mutations, A and B (grey), individually increase the measured quantitative phenotype (gene expression, for example) compared to the wildtype. In this study, we use the multiplicative expectation of how the phenotypic effects of two mutations contribute to the double mutant phenotype, according to which epistasis = $f_{m12} / (f_{m1}f_{m2})$, where $f_{m12}$ is the relative fluorescence of a double mutant ($m_{12}$), and $f_{m1}$ and $f_{m2}$ the relative fluorescence of the two corresponding single mutants ($m_1$ and $m_2$), respectively. An alternative to the multiplicative assumption would be the additive one, in which the effect of the double mutant in the absence of epistasis

*Figure 1 continued on next page*

the effects of point mutations in the *cis*-element on the binding properties of RNAP and TFs allows for the construction of a realistic model of transcription initiation (***Bintu et al., 2005a***; ***Kinney et al., 2010***), while providing a measurable and relevant phenotype - gene expression level - for the analysis of epistasis.

## Results

Here we studied epistasis between point mutations in the canonical lambda bacteriophage CRE (***Ptashne, 2011***) (***Figure 2***). We employ a fluorescent reporter protein that is under the control of the strong lambda promoter $P_R$ (***Figure 2a***), which is fully repressed by an inducible TF, CI (***Figure 2b***). RNAP and CI have overlapping binding sites in this CRE, and hence compete for binding. We created a library of 141 random double mutants in the CRE, with all their corresponding single mutants (***Supplementary file 1***). This design allows us to calculate epistasis between the mutations in the *cis*-regulatory element in two environments: in the absence of CI, when only RNAP determines expression; and in the presence of CI when the two proteins compete for binding.

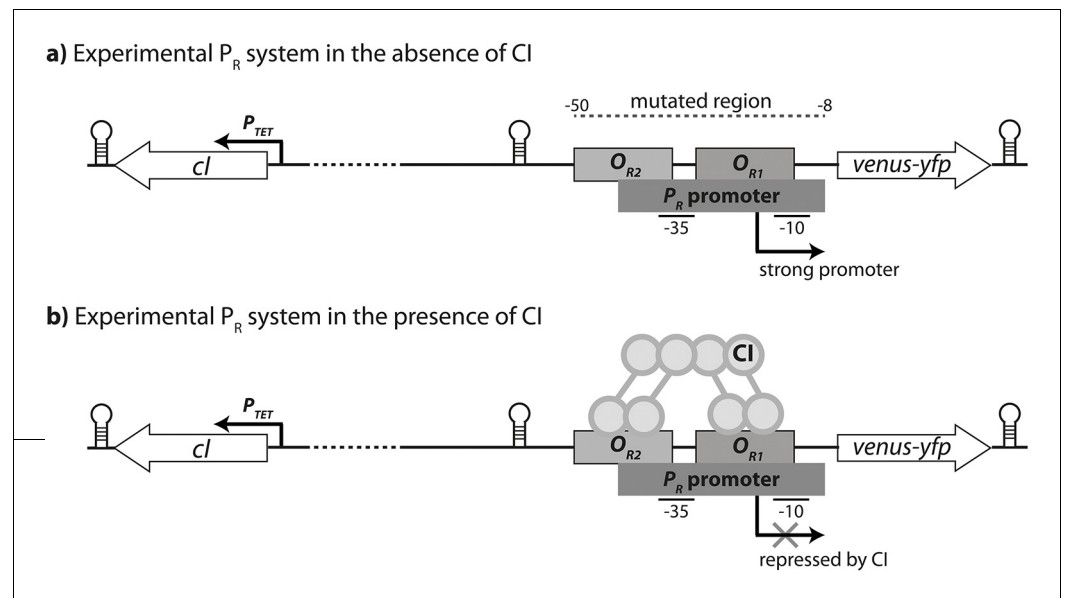

**Figure 2.** Experimental system. The $P_R$ promoter system used in the empirical measurements consists of a strong lambda phage $P_R$ promoter (RNAP-binding site) and two CI operator sites (transcription factor binding sites $O_{R1}$ and $O_{R2}$), which control the expression of a *venus-yfp* reporter gene. *cl* is encoded on the opposite strand, separated by a terminator and 500 bp of random sequence, and under the control of an inducible promoter $P_{TET}$. Both *venus-yfp* and *cl* genes are followed by a terminator sequence. (**a**) In the absence of CI, the promoter is fully expressed. (**b**) CI binds cooperatively to two operators in order to repress the promoter.

## Most double mutants change the sign of epistasis between the two environments

Throughout we assume a multiplicative model of epistasis, which defines epistasis as a deviation of the observed double mutant expression level (relative to the wildtype) from the product of the relative single mutant expression levels (*Phillips, 2008*). It should be noted that there is no *a priori* expectation for the sign of epistasis, even if most mutations are deleterious: epistasis denotes only deviations from the expected phenotype of the double mutant, and can be either positive or negative (*Figure 1*). First, we measured expression levels in the absence of CI (*Figure 3—figure supplement 1a*, *Figure 3—figure supplement 2a*). We observe that the majority of double mutants are in negative epistasis (*Figure 3a*) — the observed double mutant expression level is lower than the multiplicative expectation based on single mutant expression levels (Pearson's $\chi^2_{1,112}$=43.82, p<0.0001). Specifically, we observe negative epistasis in 83% of 113 mutants that display statistically significant epistasis, while 28 double mutants do not display significant epistasis (*Figure 3a*, *Figure 3—source data 1*).

Next, we estimated epistasis at high CI concentration, when gene expression depends on the competitive binding between RNAP and CI (*Figure 3b*, *Figure 3—figure supplement 1b*, *Figure 3—figure supplement 2b*, *Figure 3—source data 1*). In a repressible promoter, the effects of mutations on the binding of the two proteins have opposite effects on gene expression — a reduction in RNAP binding leads to a decrease in gene expression, while a reduction in CI binding leads to higher expression levels. By comparing epistasis between two environments – absence of CI and high CI concentration – we find that the 141 tested random double mutants show a strong dependence on the environment (ANOVA testing for a GxGxE interaction: $F_{1,280}$ = 21.77; p<0.0001), in line with previous observations in another bacterial regulatory system (*Lagator et al., 2016*). Interestingly, 58% of double mutants display a change in the sign of epistasis between the two environments (*Figure 4*). Especially prevalent is a switch from negative epistasis in the absence of CI, to positive epistasis in its presence (*Figure 4*). Strikingly, the proportion of double mutants exhibiting reciprocal sign epistasis (when the sign of the effect of each mutation changes in the presence of the other mutation) is greater in the presence (66%) than in the absence (8%) of CI (*Supplementary file 2*). This difference likely arises from the molecular architecture of a repressible strong promoter. Mutations affect the binding of both DNA-binding proteins, but in the presence of CI the effect on the binding of RNAP is only unmasked when CI does not fully bind, a scenario that is more likely in the presence of two mutations.

## Generic model of a simple CRE

In order to understand these observations, we created a model of gene regulation that relies on statistical thermodynamical assumptions to model the initiation of transcription, originally developed to describe gene regulation by the lambda bacteriophage repressor CI (*Ackers et al., 1982*). Importantly, our model is generic, as it does not consider the details of any specific transcription factors involved in regulation. Instead, we model competitive binding between two generic transcription factors that share a single binding site (*Figure 5a*). The binding of one of these TFs leads to an increase in the gene expression level, in a manner similar to the function of a typical RNAP or an activator. The other is a repressor molecule, the binding of which has a negative effect on gene expression level, achieved by blocking access of the activator to its cognate binding site. In order to draw a parallel to our experimental system, we refer to these two TFs in the generic model as 'RNAP' and 'repressor', without actually relying on any specific properties of the two molecules, such as CI dimerization, or cooperative binding of CI dimers to multiple operator sites.

In the thermodynamic model of transcription, each DNA-binding protein is assigned a binding energy ($E_i$) to an arbitrary stretch of DNA. In our formulation, we assume that each position along the single DNA-binding site under consideration contributes additively to the global free binding energy — an assumption found to be accurate at least for a few mutations away from a reference sequence (*Vilar, 2010*). These energy contributions can be determined experimentally (*Kinney et al., 2010*) and are typically represented in the form of an energy matrix. Given a set of DNA-binding proteins (specifically, their energy matrices) and a promoter sequence, a Boltzmann weight can be assigned to any configuration of these proteins on the promoter. The Boltzmann weight is proportional to the probability of finding the system in each of the possible configurations.

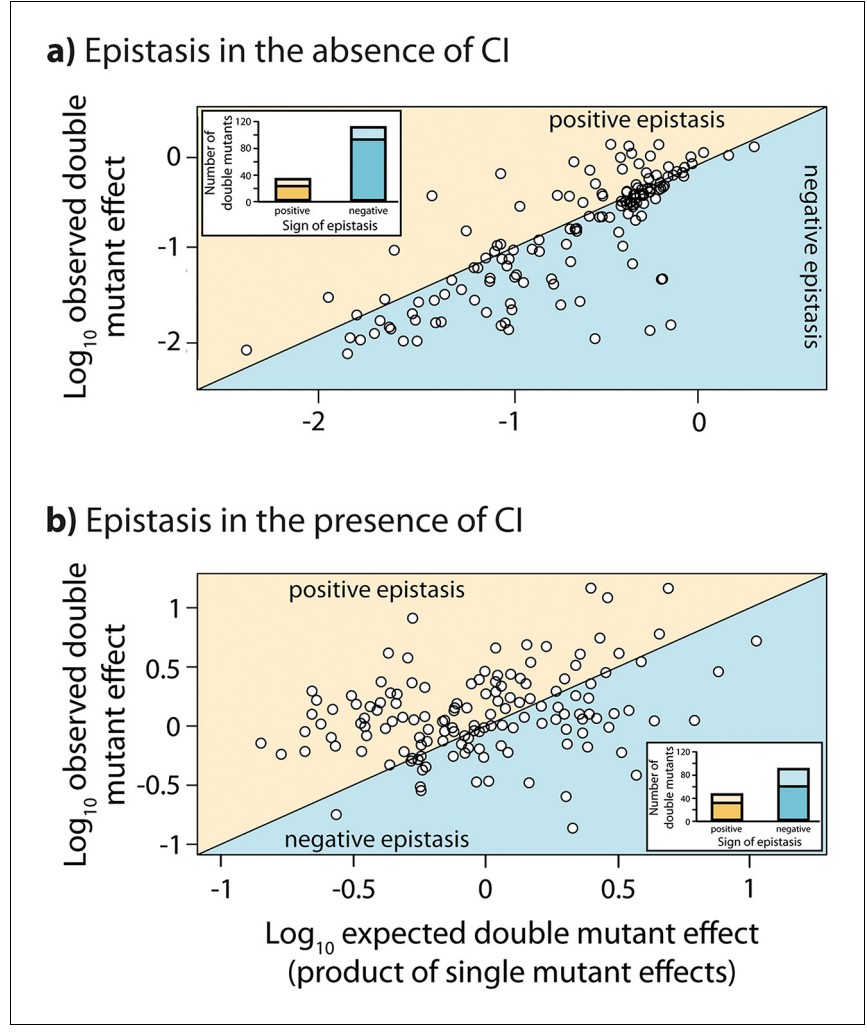

**Figure 3.** Epistasis in the absence and in the presence of CI. Points show $\log_{10}$ of expected versus $\log_{10}$ of observed double mutant effects (each relative to wildtype fluorescence) for all 141 double mutants, in the (a) absence; and (b) presence of the CI repressor. The solid line represents no epistasis (expected equal to the observed double mutant expression). Six replicates of each mutant were measured. Bar charts show total number of double mutants exhibiting positive (orange) and negative (blue) epistasis, while the darker areas represent the number that are significantly different from the null expectation of the model (no epistasis). The data presented in this figure can be found in *Figure 3—figure supplement 1*, *Figure 3—figure supplement 2*, and *Figure 3— source data 1*.

The following source data and figure supplements are available for figure 3:

**Source data 1.** Fluorescence measurements of single and double mutants, and the calculated values for epistasis for the random mutant library.

**Figure supplement 1.** Relative fluorescence of single mutants.

**Figure supplement 2.** Relative fluorescence of double mutants.

By assigning a Boltzmann weight to all configurations, one can calculate the probability of finding the system in a particular state (a set of configurations sharing a common property). Specifically, one can calculate the probability of finding the system in a configuration that leads to the initiation of transcription (*Figure 5a*).

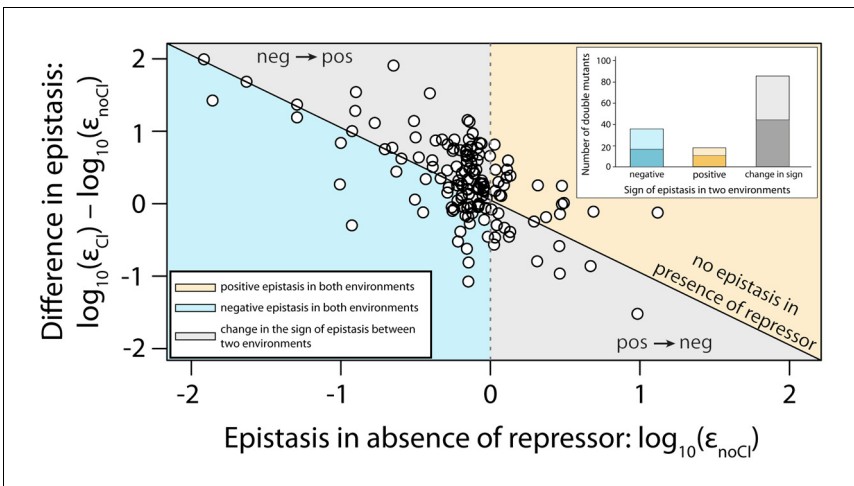

**Figure 4.** Sign of epistasis changes with the environment for most double mutants. Points show the $\log_{10}$ value of epistasis in the absence of repressor, and the difference in the $\log_{10}$ value of epistasis in the presence and the absence of repressor: $\log_{10}(\epsilon_{CI}) - \log_{10}(\epsilon_{noCI})$, for all 141 double mutants. Points above the solid diagonal line exhibit positive, while points below exhibit negative epistasis in the presence of the CI repressor. Most mutants have a different sign of epistasis between the two environments (gray area). Bar chart shows total number of double mutants that are always in positive (orange) or in negative (blue) epistasis, and the total number that changes sign between the two environments (gray). The darker areas in the bars represent the number that are significantly different from the null expectation of the model (no epistasis) in both environments. Six replicates of each mutant were measured. The data presented in this figure is calculated from *Figure 3—source data 1*.

In our generic model, we consider only a single binding site to which 'repressor' and 'RNAP' compete for binding. Note that the model does not make any assumptions about the identity of the TFs that are binding DNA and hence does not utilize any specific energy matrix. The model is, therefore, general in nature, relying only on the physical and mechanistic properties of protein-DNA binding. In such a system, three basic configurations are possible: no proteins bound to DNA, only 'RNAP' bound, or only 'repressor' bound (*Figure 5a*). Each of these states is assigned a Boltzmann weight ($Z$) based on its free binding energy $E_i$: *1*; $[P]e^{-\beta E_P}$; and $[R]e^{-\beta E_R}$, respectively, where $\beta$ is $1/k_BT$; subscript $P$ refers to 'RNAP', subscript $R$ to the 'repressor'; $[P]$ and $[R]$ to the exponential of the chemical potential for the 'RNAP' and the 'repressor' which for simplicity we equate to the concentrations of the two molecules; and $E_i$ corresponds to the change in Gibbs free energy of the reaction of the binding between protein and DNA. Assuming that the system is in thermodynamic equilibrium, we can calculate the probability of finding the system in a configuration leading to transcription ($p_{ON}$) – when RNAP is bound:

$$p_{ON} = \frac{[P]e^{-\beta E_P}}{1 + [P]e^{-\beta E_P} + [R]e^{-\beta E_R}}$$

The phenotype of a mutant is obtained by calculating $p_{ON}$ for a free energy $E'_i = E_i + \Delta$, where $\Delta$ represents the effect of the mutation on the binding of the protein to the sequence. The energies of single mutants and double mutants are $E_P^{m_1} = E_P + p_1$ and $E_R^{m_1} = E_R + p_1$; and $E_P^{m_2} = E_P + p_2$ and $E_R^{m_2} = E_R + p_2$; and $E_P^{m_{12}} = E_P + p_1 + p_2$ and $E_R^{m_{12}} = E_R + p_1 + p_2$, respectively, where $p_i$ stands for the effect of mutation $i$ on the binding of 'RNAP' and $r_i$ for the effect on 'repressor' binding. From these measures of the mutational effects, we calculated epistasis against a multiplicative model, in the same manner as done for the experimental measurements:

$$p_{ON}^{m_{12}} = \epsilon p_{ON}^{WT} \frac{p_{ON}^{m_1} p_{ON}^{m_2}}{p_{ON}^{WT} p_{ON}^{WT}}$$

With the generic model, we ask only about the sign of epistasis and say that it is positive when $\epsilon > 1$ and negative when $\epsilon < 1$. The generic model cannot predict the magnitude of epistasis in any

**a)** Possible occupancy states in the thermodynamic model of gene regulation by binding-site competition

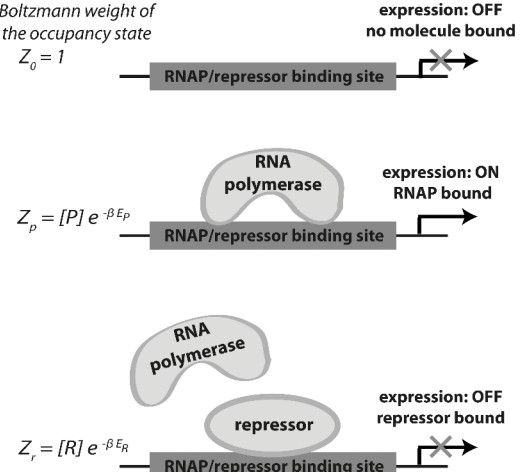

**b)** Predictions of epistasis from the thermodynamic model

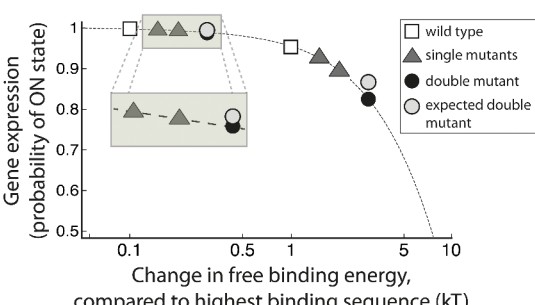

**c)** Thermodynamic model predicts that sign of epistasis depends on sign of individual mutation effects

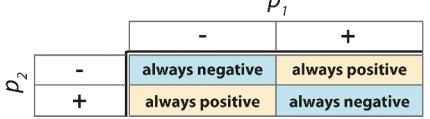

**d)** Expression when single mutants affect only RNAP or repressor

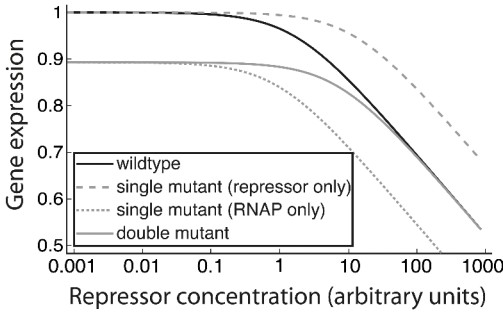

**e)** Epistasis when single mutants affect only RNAP or repressor

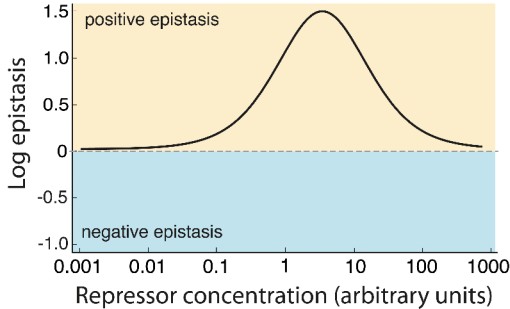

**f)** Expression when single mutants affect both RNAP and repressor

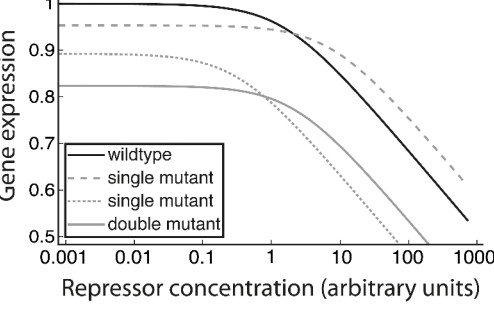

**g)** Epistasis when single mutants affect both RNAP and repressor

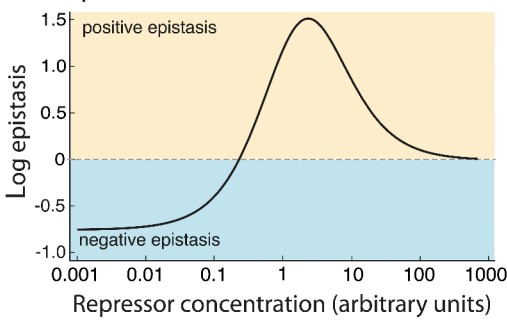

**Figure 5.** Overview of the generic model. The theoretical approach used in this study, originally developed to describe gene regulation by the lambda bacteriophage repressor CI (**Ackers et al., 1982**), relies on statistical thermodynamics assumptions to model initiation of transcription. (a) In this framework, each DNA-binding protein is assigned a binding energy ($E_i$) to an arbitrary stretch of DNA. Given a set of DNA-binding proteins (a generic RNAP-like and a generic repressor-like TF, in this case) and a promoter sequence, a Boltzmann weight can be assigned to any configuration of these TFs on the promoter. By assigning a Boltzmann weight to all configurations, one can calculate the probability of finding the system in a configuration that leads to the initiation of transcription. (b) When considering only the binding of a single protein to DNA (for example 'RNAP' only), if mutations

*Figure 5 continued on next page*

*Figure 5 continued*

have a negative effect on protein-DNA binding, the model predicts negative epistasis between them in terms of expression. This prediction arises from the non-linear relationship between binding energy and gene expression $p_{on}$ (dotted line). In this illustration, we show a relative change in binding energy compared to the sequence with highest possible binding, in kT. (**c**) By generalizing the properties of the relationship between binding and gene expression, we conclude that the sign of epistasis depends only on the sign of individual mutation effects ($p_1$ and $p_2$) upon binding. When both 'RNAP' and 'repressor' are present in the system, epistasis depends on the 'repressor' concentration and the magnitude of single mutation effects on 'RNAP' and 'repressor' binding (**d,e,f,g**). (**d**) One point mutation negatively affects only 'RNAP' binding, while the other only 'repressor' binding. (**e**) Under such circumstances, the system shows no epistasis at low 'repressor' concentrations, but is in positive epistasis when 'repressor' concentration increases. Finally, at very high repressor concentrations, epistasis approaches 0. (**f**) Point mutations negatively affect both 'RNAP' and 'repressor' binding. (**g**) Under such conditions, epistasis changes the sign from negative to positive as repressor concentration increases.

particular biological system without accounting for the underlying energy matrices and intracellular concentrations of relevant TFs. As the model does not account for the details of any specific regulatory system, it considers only the direct, primary effects of a mutation on binding affinity (*Bintu et al., 2005a*), and does not consider any potential interactions arising from secondary effects, namely the effects of a mutation on the structure of DNA (*Rajkumar et al., 2013*), accessibility to the binding sites (*Levo and Segal, 2014*), protein cooperativity (*Todeschini et al., 2014*), looping (*Levine et al., 2014*), or any other potential regulatory structures.

## The sign of epistasis can be predicted from first principles

Using the generic model, we first studied the effects of mutations only on 'RNAP' binding (in the absence of 'repressor'), and found that epistasis depends only on the sign of individual mutation effects (*Figure 5*). Our model predicts that if mutations have the same sign, they are always in negative epistasis. This prediction arises from the non-linear relationship between binding energy and expression $p_{on}$ (*Figure 5b*). Namely, when repressor concentration goes to zero, epistasis is negative only when $e^{-p_1} + e^{-p_2} - p_1 - p_2$ - a condition satisfied only when $p_1$ and $p_2$ have the same sign. Conversely, when the two mutations have a different sign, they will always be in positive epistasis. In general, the physical properties of the relationship between binding and gene expression indicate that the sign of epistasis for any given TF depends only on the sign of individual mutation effects ($p_1$ and $p_2$) upon binding (*Figure 5c*).

Experimental observations do not significantly differ from these predictions for the sign of epistasis ($\chi^2_{1,112}=3.64$, p=0.056), as 96 of the 113 double mutants (85%) that are in significant epistasis in the absence of CI conform to model predictions. Experimental deviations from the generic model predictions (i.e. displaying positive epistasis when both mutations have the same sign) could be due to the secondary effects of mutations, as they could affect the general context of RNAP binding (*Rajkumar et al., 2013*), or the ability of CI to bind cooperatively (*Stayrook et al., 2008*).

The model also describes patterns of epistasis in the presence of a repressor. By assuming that every point mutation affects the binding of both 'RNAP' and 'repressor', we find that the environmentally dependent change in the sign of epistasis depends on the concentrations of 'RNAP' and 'repressor', as well as the sign and relative magnitude of individual mutation effects (*Table 1—source data 1*). At high 'repressor' concentrations, effects of mutations on 'repressor' binding dominate over their effects on 'RNAP binding'. In these environments, the sign of epistasis depends only on the sign of individual mutation effects on 'repressor' binding.

In general, assuming that 'RNAP' concentration stays relatively constant (*Raser and O'Shea, 2005*) allows us to derive how the sign of epistasis depends on repressor concentration (*Table 1*). When one point mutation negatively affects only 'RNAP' binding, while the other only 'repressor' binding (*Figure 5d*), the system does not exhibit any epistasis when 'repressor' concentration is very low, as only one of the mutations affects 'RNAP' binding (*Figure 5e*). As 'repressor' concentration increases, the system is in positive epistasis. Finally, at very high 'repressor' concentrations, which are probably not biologically relevant, epistasis approaches 0 as the 'repressor' binds too strongly. When point mutations negatively affect both 'RNAP' and 'repressor' binding (*Figure 5f*), epistasis changes the sign from negative to positive as 'repressor' concentration increases (*Figure 5g*).

To intuit this finding, consider two mutations that reduce binding of both 'RNAP' and 'repressor'. In the absence of 'repressor', when only 'RNAP' is present, epistasis will be negative because of the

**Table 1.** Sign of epistasis in a simple CRE depends on the environment and the sign of individual mutation effects. We consider two environments, one without repressor when mutations affect only RNAP binding, and the other with high repressor concentration. In the first environment, sign of epistasis is determined only by the sign of individual mutation effects on RNAP binding, while in the second environment it is the sign of individual mutation effect on the repressor that matters. For each mutation, the signs ('+' and '-') represent the sign of its effect on the binding of RNAP ($p$) and repressor ($r$), respectively. 'neg -> pos' and 'pos -> neg' represent combinations that display transitions from negative to positive, or positive to negative epistasis, respectively. Certain combinations of mutations are always in negative or always in positive epistasis. The extended version of this table, which does not assume a constant 'RNAP' concentration in the cell, is provided in **Table 1—source data 1**.

|  |  | $p_1 r_1$ | | | |
|---|---|---|---|---|---|
|  |  | −− | +− | −+ | ++ |
|  | ++ | pos→neg<br>neg→pos | neg→pos | pos→neg | neg→pos |
| $p_2 r_2$ | −+ | pos→neg | always positive | always negative |  |
|  | +− | pos→neg | always negative |  |  |
|  | −− | neg→pos |  |  |  |

Source data 1. General conditions for the sign of epistasis in two environments.

negative curvature of the relationship between expression and binding energy (*Figure 5b*). But, in the presence of 'repressor', it is the relative magnitude of individual mutation effects that will determine the sign of epistasis. This is because mutations that weaken 'repressor' binding increase expression. If the mutation effects are larger on 'RNAP', then the negative epistasis on expression arising from 'RNAP' will dominate. When the mutations have a greater effect on 'repressor' binding, then negative epistasis on 'repressor' binding will dominate and lead to positive epistasis on expression, and hence to a dependence on the environment. At high 'repressor' concentration, only the sign of the effects of mutations on 'repressor' binding will determine the sign of epistasis. As most experimentally tested mutations reduce both RNAP and CI binding, our model explains the observation that most double mutants change the sign of epistasis between the two environments (*Figure 4*).

## Independent validation of the generic model predictions

The experimental data from the random mutant library (*Figures 3* and *4*) shows that the patterns of epistasis between two environments follow the generic model predictions, specifically that epistasis switches sign between environments in many mutants. However, our experimental design, where we only measure gene expression levels, does not allow us to identify the effects of a mutation on CI binding alone. For example, if a mutation decreases gene expression level in the presence of CI, we cannot know if it decreases RNAP binding, increases CI binding, or both. This prevents a more thorough verification of the generic model. In order to independently experimentally validate the generic model predictions (*Table 1*), it is necessary to know the effects of CRE mutations on RNAP and CI. To obtain this information, we used the experimentally determined energy matrices for RNAP (*Kinney et al., 2010*) and CI (*Sarai and Takeda, 1989*), and utilized it to create five random double mutants for each possible combination of single mutation effects shown in *Table 1*. Due to the high specificity of binding of both RNAP and CI, we could not identify point mutations that simultaneously improved the binding of both (*Supplementary file 3*). Therefore, we validate the model by measuring epistasis in 30 double mutants (five for each of the six possible combinations of single mutant effects) in the two environments. We find no difference between the predicted and experimental estimates of the sign of epistasis and its dependence on the two experimental environments

(Pearson's $\chi^2_{2,30}$=0.68; p=0.72) (*Figure 6*). As such, the predictions about the sign of epistasis that arise from the generic model (*Table 1*) hold true in our experimental system.

Furthermore, we tested if a simple thermodynamic model that incorporates the two energy matrices (*Sarai and Takeda, 1989*; *Kinney et al., 2010*) can predict not only the sign, but also the magnitude of epistasis in the two environments. Because such a model depends on the concentrations of RNAP and CI, we estimated the values for these parameters so as to maximize the correlation between model predictions and empirical values of epistasis. When we excluded those double mutants which did not empirically exhibit significant epistasis, we found a significant fit between experimental measurements and model predictions of the magnitude of epistasis in the absence ($F_{1,15}$ = 9.86; p<0.01) and in the presence of CI ($F_{1,15}$ = 4.59; p<0.05) (*Figure 6—figure supplement 1*). As such, the model predicts not only the general patterns of epistasis (sign), but is also reasonably accurate at predicting its magnitude, which is remarkable since the model does not consider detailed molecular aspects of the experimental system, such as CI dimerization or cooperativity.

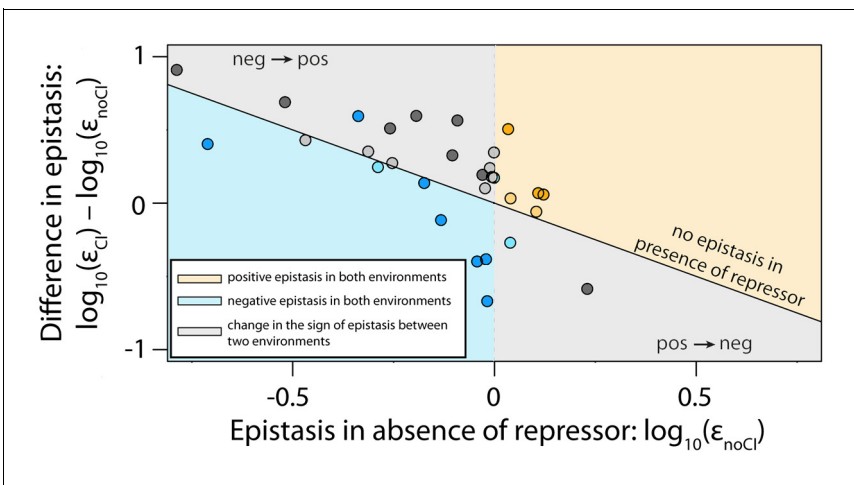

**Figure 6.** The thermodynamic model accurately predicts sign of epistasis and its environment-dependence. In order to conduct an independent test of the assumptions of the generic model, we expanded the generic model to include specific information about the two TFs relevant to the experimental system – namely, the energy matrices for RNAP (*Kinney et al., 2010*) and CI (*Sarai and Takeda, 1989*). We could not use the 141 random mutants to validate the model, as most of them contained mutations that were in the regions of the CRE that were poorly characterized by the energy matrices. Therefore, using the energy matrices, we had to create a new library consisting of five random double mutants for each category from *Table 1*. As we could not identify any single point mutations that simultaneously improved the binding of both RNAP and repressor, we tested if empirical measurements of epistasis conformed to model predictions in 30 mutants. The model predictions of the sign of epistasis and its environment dependence were based only on the sign of individual mutation effects on RNAP and repressor binding. The location of points corresponds to the experimental measurement of epistasis for each mutant, while the color indicates the model prediction: (i) blue - double mutants predicted to be in negative epistasis both in the absence and in the presence of the repressor CI; (ii) orange - double mutants that are always in positive epistasis; (iii) grey - double mutants predicted to change the sign of epistasis in the two environments. The color intensity indicates significance – lighter shades represent non-significant, darker shades represent significant epistasis in both environments (see 'Empirical verification of the thermodynamic model' section in Online Methods). Six replicates of each mutant were measured. The data underlying this figure is presented in *Figure 6—source data 1*. The quantitative test of how well the thermodynamic model predicts the magnitude of epistasis in this dataset is presented in *Figure 6—figure supplement 1*.

The following source data and figure supplement are available for figure 6:

**Source data 1.** Fluorescence measurements of single and double mutants, and the calculated values for epistasis for the validation mutant library.

**Figure supplement 1.** The thermodynamic model predicts the magnitude of epistasis.

## Discussion

The theory we present here, which is based on mechanistic properties of protein-DNA binding without accounting for any details of the molecular system studied, provides an accurate prediction of the sign of epistasis and its environmental dependence for a repressible promoter system - the most common form of gene regulation in *E. coli* (~40% of all regulated genes [*Salgado et al., 2013*]). Furthermore, the fact that we use a generic model with no reference to any particular empirical measures means that our results are derived from first principles. As such, the presented results should hold as long as the effects of mutations on gene expression are mainly driven by their direct impact on TF-DNA binding, as represented by the energy matrix for a given TF. Under such conditions, the thermodynamic model, rather than the multiplicative (or additive) expectation, provides a meaningful null model for the sign of epistasis in CREs.

The sign of the deviations from a multiplicative expectation can have important evolutionary consequences, such as for the evolution of sex (*Otto and Lenormand, 2002*) or the maintenance of genetic variation (*Charlesworth et al., 1995*). A particularly important pattern of epistasis is sign epistasis, where the sign of the effect of a particular substitution depends on the genetic background. Sign epistasis can lead to the existence of multiple optima (local peaks). In the system we analyze here, sign epistasis cannot exist in the absence of a repressor, since there is an optimum binding site sequence and the effects of mutations have a definite sign toward this optimal sequence. In the presence of a repressor, however, sign epistasis is possible (*Poelwijk et al., 2011*). Furthermore, we show that the sign of epistasis very often reverses between environments. This phenomenon, previously observed in a different system (*de Vos et al., 2013*; *Lagator et al., 2016*), could alleviate constraints coming from the existence of multiple peaks in a particular environment. The thermodynamic model provides a mechanistic basis for this observation: RNAP and repressor have opposite effects on gene expression and this, when combined with the specific shape of response induced by the thermodynamic model, can lead to the environmental dependence of the sign of epistasis.

Our results concern the combined effect of mutations (epistasis) on phenotype, as opposed to fitness. Phenotypes logically precede fitness and even though it could be argued that fitness is 'what matters' for evolution, since mutations spread in part based on their fitness effects, determining the fitness effects of mutations depends on the environment which may or may not be representative of 'natural' conditions. Moreover, knowledge about one environment is hardly informative about the fitness patterns in a novel environment. Our results allow for the prediction of patterns of phenotypic epistasis across different environmental conditions, independent of the selection pressures applied to this phenotype. The evolutionary consequences of these patterns of epistasis can then be inferred from the knowledge (or assumptions) of how selection is acting on this phenotype, or in other words, how the phenotype maps onto fitness.

In order to predict the sign of epistasis in a regulatory system, the thermodynamic model accounts for the underlying physical mechanisms that impose constraints on the genotype-phenotype map under consideration. Incorporating details of physical and molecular mechanisms into models of more complex regulatory elements, as well as coding sequences (*Dean and Thornton, 2007*; *Li et al., 2016*), can elucidate how epistasis impacts genotype-phenotype maps and their dynamic properties across environments, helping us to understand the environmental dependence of fitness landscapes.

## Materials and methods

### Gene regulation in the $P_R$ promoter system

We developed a system based on the *right* regulatory element of the lambda phage ($P_R$), in which we decoupled the *cis*- and *trans*-regulatory elements (*Figure 2*) (*Johnson et al., 1981*). A *Venus-yfp* gene (*Nagai et al., 2002*) is placed under the control of the *cis*-regulatory region containing the $P_R$ promoter with two lambda repressor CI-binding sites ($O_{R1}$ and $O_{R2}$). The transcription factor CI represses the $P_R$ promoter by direct binding-site competition with RNAP. Separated by 500 random base pairs and on the opposite DNA strand, we placed the *cl* repressor gene under the control of a $P_{TET}$ promoter (*Lutz and Bujard, 1997*), followed by a TL17 terminator sequence. Thus, concentration of CI transcription factor in the cell was under external control, achieved by addition of the

inducer anhydrotetracycline (aTc). The entire cassette was inserted into the low-copy number plasmid pZS* carrying kanamycin resistance gene (*Lutz and Bujard, 1997*).

## Random mutant library

We created a library of random single and double mutants in the 43 bp *cis*-regulatory element (consisting of the RNAP binding site and the two CI operator sites $O_{R1}$ and $O_{R2}$) using the GeneMorph II™ random mutagenesis kit (Agilent Technologies, Santa Clara, US). PCR products of mutagenesis reactions were ligated into the wildtype plasmid and inserted into a modified *Escherichia coli* K12 strain MG1655 chromosomally expressing *tetR* gene from a $PN_{25}$ promoter. We sequenced ~500 colonies in order to create a library of 141 double mutants for which both corresponding single mutants were also identified (*Supplementary file 1*). We identified, in total, 89 mutants carrying only a single point mutation. Four single and four double mutants from the library were randomly selected and the whole plasmid sequenced to confirm that during library construction no mutations were found outside the target regulatory region.

We measured fluorescence for each single and double mutant, as well as the wildtype $P_R$ promoter system, both in the presence and in the absence of the inducer aTc. Six replicates of each mutant in the library were grown overnight in M9 media, supplemented with 0.1% casamino acids, 0.2% glucose, 30 µg/ml kanamycin, either without or with 15 ng/ml aTc. Presence or absence of aTc determined the two experimental environments. Overnight cultures were diluted 1000X, grown to $OD_{600}$ of approximately 0.05, and their fluorescence measured in Bio-Tek Synergy H1 platereader. The measured fluorescence was first corrected for the autofluorescence of the media, and then normalized by the wildtype fluorescence. All replicate measurements were randomized across multiple 96-well plates. All replicates were biological, having been kept separate from each other from the moment that the mutant was cloned and identified through sequencing. Six replicates of each mutant were measured as prior experience with similar datasets in the lab has shown it sufficient to detect meaningful differences between mutants.

## Statistical analyses

By using a multiplicative model of epistasis, we calculated epistasis relative to the wildtype as $\epsilon = f_{m12}/(f_{m1}f_{m2})$, where $f_{m12}$ is the relative fluorescence of a double mutant (m12), and $f_{m1}$ and $f_{m2}$ the relative fluorescence of the two corresponding single mutants (m$_1$ and m$_2$), respectively. In order to determine statistically which double mutants exhibit epistasis (i.e. ε not equal 1), we conducted a series of FDR-corrected *t*-tests. The errors were calculated based on six replicates, using error propagation to account for the variance due to normalization by the wildtype. Variance is not significantly different between measured mutants (*Figure 3—figure supplement 1*; *Figure 3—figure supplement 2*). We performed a Pearson's *chi*-squared test to determine if double mutants had a tendency toward negative epistasis. We asked whether epistasis depended on the environment (defined as presence or absence of the repressor) by testing for a genotype x genotype x environment (GxGxE) interaction using ANOVA. We also tested if the experimental observations of the sign of epistasis in the absence of CI repressor corresponded to model predictions. To do that, we used the experimental measurements of the sign of single mutation effects to predict the sign of epistasis (if both mutations had the same sign then epistasis was predicted to be negative, if they differed in sign, it was predicted as positive). Then we compared the predicted distribution of the sign of epistasis to the experimental estimates using a *chi*-squared test, limiting the test to only those double mutants that experimentally exhibited significant epistasis. For all tests, data met the assumptions, and variance between groups was not significantly different.

## Generic model of gene regulation with binding site competition between RNAP and repressor

The model is based on previous thermodynamic approaches (*Bintu et al., 2005a*, *2005b*; *Hermsen et al., 2006*). These models consider all possible promoter occupancy states and assign a Boltzmann weight to each state. The probability of any microstate (promoter configurations) is given by Boltzmann weights $W_i = e^{-\beta(E_i - N\mu)}$, where $E_i$ is the Gibbs free energy of the configuration, $N$ is the number of TF molecules, $\beta$ is $1/K_B T$, and $\mu$ represents the chemical potential. $P_{on}$ can

then be calculated as the normalized sum of all configurations conducive to the initiation of transcription:

$$p_{ON} = \frac{\sum_{i \in \oplus} w_i}{\sum_i w_i}$$

where the first summation is over the all configurations conducive to transcription, whereas the second is over all configurations.

In our model, we consider a scenario in which an activator (such as RNAP) competes with a repressor for access to its binding site. We consider only three possible promoter configurations: the one where neither of the two proteins is bound, the one in which a 'repressor' prevents 'RNAP' from accessing its binding site, and the one in which' RNAP' is bound to its binding site, thereby able to initiate transcription. Under these assumptions, the probability of initiation of transcription is:

$$p_{ON} = \frac{[P]e^{-\beta E_P}}{1 + [P]e^{-\beta E_P} + [R]e^{-\beta E_R}}$$

where $[P]$ and $[R]$ represent the exponential of the chemical potential for the 'RNAP' and the 'repressor', respectively; and subscripts $P$ and $R$ represent 'RNAP' and 'repressor', respectively. Throughout, we measure free energies in natural units such that $\beta = 1$.

We assume that mutations simultaneously affect the binding of both 'RNAP' and 'repressor' to the DNA binding site. We denote the free energies of both 'RNAP' and 'repressor' binding to DNA by $E_P$ and $E_R$, respectively. We model the effect of mutations by perturbing these energies by an additive factor. The energies of single mutants and double mutants are then $E_P^{m_1} = E_P + p_1$ and $E_R^{m_1} = E_R + p_1$; and $E_P^{m_2} = E_P + p_2$ and $E_R^{m_2} = E_R + p_2$; and $E_P^{m_{12}} = E_P + p_1 + p_2$ and $E_R^{m_{12}} = E_R + p_1 + p_2$, respectively,

We calculate epistasis against a multiplicative model for the effect of mutations on $p_{ON}$:

$$p_{ON}^{m_{12}} = \varepsilon p_{ON}^{WT} \frac{p_{ON}^{m_1} p_{ON}^{m_2}}{p_{ON}^{WT} p_{ON}^{WT}}$$

and so epistasis is measured by:

$$\varepsilon = \frac{p_{ON}^{WT} p_{ON}^{m_{12}}}{p_{ON}^{m_1} p_{ON}^{m_2}} = \frac{(1 + Ae^{-r_1} + Be^{-p_1})(1 + Ae^{-r_2} + Be^{-p_2})}{(1 + A + B)(1 + Ae^{-r_1-r_2} + Be^{-p_1-p_2})}$$

where $A = [R]e^{-E_R}$ and $A = [P]e^{-E_P}$. We say that epistasis is positive when $\varepsilon > 1$ and negative when $\varepsilon < 1$. We then find the conditions for which epistasis is positive in the presence $(A > 0)$ or absence $(A = 0)$ of repressor.

## Empirical verification of the generic model

In order to empirically test the predictions of the generic model on the relationship between sign of individual mutations and the sign of epistasis in two environments, we aimed to select five random double mutants from each category from *Table 1*. Effects of mutations on RNAP and on CI were obtained from the experimentally determined energy matrices of RNAP (*Kinney et al., 2010*) and CI (*Sarai and Takeda, 1989*) binding. We could not validate the model from the random mutant library, as the majority of mutants fell in regions that are poorly described by the energy matrices. For this reason, we aimed to create this new library. As the $P_R$ promoter is very strong, finding double mutants where both mutations improved expression was not possible. Hence, we selected five double mutants from six categories (*Supplementary file 3*), and synthesized them, as well as their corresponding single mutants, using annealed oligonucleotide overlap cloning. We measured fluorescence of these mutants and calculated epistasis in the same manner as described for the random mutant library, and we asked if the epistasis for each double mutant was different from the null-expectation in the manner described in section 'Statistical analyses'. We used Pearson's *chi*-square test to determine if the environmental-dependence of the sign of epistasis in the experimental measurements differs from model predictions.

In order to test whether the thermodynamic model can also predict the magnitude of epistasis, we incorporated the energy matrices for RNAP (*Kinney et al., 2010*) and CI (*Sarai and Takeda,*

*1989*) into the generic model. As the energy matrix for RNAP contained one additional position in the spacer region between $-10$ and $-35$ sites compared to the experimental $P_R$ system, we eliminated one position in that region that had lowest impact on overall RNAP binding. In the manner described above, we modeled epistasis in those mutants from the 30-mutant validation library that exhibited significant epistasis. As the thermodynamic model depends on the concentrations of RNAP and CI, we estimated the values for these parameters so as to maximize the correlation between model predictions and empirical values of epistasis. In order to estimate how well the model predicted the magnitude of epistasis, we fitted a linear regression between experimental measurements of epistasis and the model predictions, both in the absence and in the presence of CI.

## Acknowledgements

We thank H Acar, S Sarikas, and G Tkacik for discussion and useful comments on the manuscript. This work was supported by the People Programme (Marie Curie Actions) of the European Union's Seventh Framework Programme (FP7/2007-2013) under REA grant agreement n° [291734] to ML, has received funding from the European Union's Seventh Framework Programme for research, technological development and demonstration under grant agreement no 618091 (SAGE) to TP and European Research Council under the Horizon 2020 Framework Programme (FP/2007–2013) / ERC Grant Agreement n. [648440] to JPB.

## Additional information

### Funding

| Funder | Grant reference number | Author |
|---|---|---|
| Seventh Framework Programme | FP7/2007-2013 | Mato Lagator |
| Seventh Framework Programme | 618091 (SAGE) | Tiago Paixao |
| Horizon 2020 Framework Programme | FP/2007-2013 | Jonathan P Bollback |

The funders had no role in study design, data collection and interpretation, or the decision to submit the work for publication.

### Author contributions

ML, Conceptualization, Data curation, Formal analysis, Investigation, Visualization, Methodology, Writing—original draft; TP, Conceptualization, Data curation, Software, Formal analysis, Funding acquisition, Investigation, Visualization, Methodology, Writing—original draft; NHB, JPB, Conceptualization, Supervision, Funding acquisition, Project administration, Writing—review and editing; CCG, Conceptualization, Supervision, Funding acquisition, Methodology, Project administration, Writing—review and editing

### Author ORCIDs

Mato Lagator, http://orcid.org/0000-0001-7847-3594
Tiago Paixão, http://orcid.org/0000-0003-2361-3953
Călin C Guet, http://orcid.org/0000-0001-6220-2052

## Additional files

### Supplementary files

• Supplementary file 1. Identity of randomly generated double mutants.

• Supplementary file 2. Types of epistasis in two environments.

• Supplementary file 3. Identity of mutants used for verification of the model.

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
