## [Decision Letter]

Thank you for submitting your article "On the Mechanistic Nature of Epistasis in a Canonical *cis*-Regulatory Element" for consideration by *eLife*. Your article has been reviewed by three peer reviewers, one of whom, Alvaro Sanchez (Reviewer #1), is a member of our Board of Reviewing Editors and the evaluation has been overseen by Aviv Regev as the Senior Editor. The following individuals involved in review of your submission have agreed to reveal their identity: Lucas B Carey (Reviewer #2).

The reviewers have discussed the reviews with one another and the Reviewing Editor has drafted this decision to help you prepare a revised submission.

Summary:

The paper addresses an interesting question, concerning the ability to predict the sign of epistasis. Non-additive (or non-multiplicative) effects of mutations on gene expression are common. The authors show that in a complex system in which both an activator (RNA Pol) and a repressor (CI) bind to the same sequence, the effect of mutations depends on both the environment (CI concentration) and on other mutations. This is an important problem in quantitative biology, and the authors have successfully resolved it for a specific system using a combination of theory and experiment.

Essential revisions:

All three referees expressed concerns regarding the presentation and a general lack of clarity of the manuscript. In order for the paper to be ready for publication, several major changes need to be made to the writing to improve its clarity and rigor. A consolidated list of required changes is below:

1) The thermodynamic model is an integral part of the paper and it needs to be described in detail in its own separate "Modeling" or "Theory" section of the paper. The authors should start the section by explaining what assumptions go into the model, explicitly stating which features of the system are being included and which excluded for the sake of simplicity. For instance, the authors should make it clear that their model ignores well known features of this system, such as explicit modeling of cI dimerization, cooperative binding of two cI dimers to OR1 and OR2, as well as any other assumptions that are currently implicit and not discussed. The authors should also explicitly discuss the limitations of their approach. For instance, if the model ignores any of these features for the sake of clarity, the authors should explicitly state how they expect this to affect a quantitative comparison between theory and experiment, and whether the model should be taken as a semi-quantitative comparison or not.

2) Along these lines, the referees were concerned about the lack of quantitative calibration of the model. Although it is understood that the model is used in a semi-quantitative fashion as a way to obtain the sign of epistasis rather than its magnitude, all referees found it would be necessary to establish how quantitatively accurate their model is, given the many assumptions it makes (as opposed to, for instance, more detailed models such as those of Ian Dodd et al. (Genes Dev 2004)). A way to do this would be to provide a quantitative comparison between their model and an independent data set. This is an approach that has been followed multiple times to calibrate thermodynamic models; See for instance the comparison between theory and experiment by Vilar & Saiz (PNAS 2005, NAR 2008), or Bintu et al. using the data from Muller-Hill et al. on the lac system (Bintu et al. 2005). One good data set where single point mutants have been studied can be found in the paper by by Sarai and Takeda, 1989. The authors should compare their theory with the results of Sarai & Takeda (or an alternative paper if the authors know of it). Note that we do not expect a perfect correlation between theory and experiment, but rather that the authors use such a comparison to establish how quantitative their model is, and whether we should just take it as a toy model that still provides some useful insights or a more serious thermodynamic model that captures the essentials of the system and provides reasonable predictive power. Either way would be fine, but the authors need to discuss this.

3) In the box (which may be instead re-purposed as a Figure that could go with the new theory section), Panel b is rather perplexing (and needs to be explained better). The authors plot Gene expression (Shouldn't this be fold-change in expression?) as a function of Ep (which, if interpreted correctly, is the free energy of binding of the polymerase). The plot is unusual because the free energy goes from +0.05 to +10 (in units of kT?). A positive free energy represents the case of repulsion between polymerase and the promoter. While technically the plot is correct, in that for a repulsive free energy, the higher the energy the lower expression would be, that is never the case and all known binding free energies between RNAP and promoters are of course negative and thus favorable (see Bintu et al., Dodd et al., and a long etc.). This point is particularly unclear in its relation with the rest of the paper, so it should be clarified.

4) A second panel to Figure 2 should be presented containing the +CI data.

5) The authors should explain in Figure 4 why they synthesized 30 new variants, 17 with statistically significant epistasis, to test the model? Why not use all 141 mutants, and use the single mutants to predict the doubles? The authors should explain their logic for doing this, particularly in relation to how quantitatively calibrated their model is.

6) The concept of sign epistasis should be better explained and motivated in the introduction. For example, when it is first introduced (in the Introduction section), it is not defined. For many readers, this will make it very difficult to understand the paper. The importance of sign epistasis is detailed in the second paragraph of the Discussion. As this is important to motivate the work, the authors should move it to the introduction.

---

## [Author Response]

*Essential revisions:*

*All three referees expressed concerns regarding the presentation and a general lack of clarity of the manuscript. In order for the paper to be ready for publication, several major changes need to be made to the writing to improve its clarity and rigor. A consolidated list of required changes is below:*

In order to improve the presentation and general clarity we made extensive changes to the manuscript. Below we explain point-by-point each of these changes. We also would like to thank the reviewers and the handling editor for the constructive input and criticism, which helped us improve the manuscript.

*1) The thermodynamic model is an integral part of the paper and it needs to be described in detail in its own separate "Modeling" or "Theory" section of the paper. The authors should start the section by explaining what assumptions go into the model, explicitly stating which features of the system are being included and which excluded for the sake of simplicity. For instance, the authors should make it clear that their model ignores well known features of this system, such as explicit modeling of cI dimerization, cooperative binding of two cI dimers to OR1 and OR2, as well as any other assumptions that are currently implicit and not discussed. The authors should also explicitly discuss the limitations of their approach. For instance, if the model ignores any of these features for the sake of clarity, the authors should explicitly state how they expect this to affect a quantitative comparison between theory and experiment, and whether the model should be taken as a semi-quantitative comparison or not.*

We made extensive changes to the manuscript by adding a modeling section titled ‘Generic model of a simple CRE’. We have also removed Box 1 and replaced it with Figure 5. Now we provide substantially more information in the main text on how the model works, what its basic assumptions are, as well as why we created a generic model (one that does not account for any specific TFs). The new Figure 5 now contains the accompanying description of how the generic model works, as well as what its main conclusions are. After explaining how the model works, we then apply it to a simple CRE, one that is akin to the experimental system used in this study, and explain what predictions arise from it regarding epistasis and its dependence on repressor concentration and individual mutation effects. To this effect, we have introduced changes to the sections: ‘The sign of epistasis can be predicted from first principles’ and ‘Independent validation of generic model predictions’. We think that this new section as well as the other changes, clarify what the model does and how it is used in this study

*2) Along these lines, the referees were concerned about the lack of quantitative calibration of the model. Although it is understood that the model is used in a semi-quantitative fashion as a way to obtain the sign of epistasis rather than its magnitude, all referees found it would be necessary to establish how quantitatively accurate their model is, given the many assumptions it makes (as opposed to, for instance, more detailed models such as those of Ian Dodd et al. (Genes Dev 2004)). A way to do this would be to provide a quantitative comparison between their model and an independent data set. This is an approach that has been followed multiple times to calibrate thermodynamic models; See for instance the comparison between theory and experiment by Vilar & Saiz (PNAS 2005, NAR 2008), or Bintu et al. using the data from Muller-Hill et al. on the lac system (Bintu et al. 2005). One good data set where single point mutants have been studied can be found in the paper by by Sarai and Takeda, 1989. The authors should compare their theory with the results of Sarai & Takeda (or an alternative paper if the authors know of it). Note that we do not expect a perfect correlation between theory and experiment, but rather that the authors use such a comparison to establish how quantitative their model is, and whether we should just take it as a toy model that still provides some useful insights or a more serious thermodynamic model that captures the essentials of the system and provides reasonable predictive power. Either way would be fine, but the authors need to discuss this.*

We tested how quantitatively accurate the thermodynamic model is in predicting our experimental data, and present the findings in Figure 6—figure supplement 1, and in the section ‘Independent validation of the generic model predictions’. In order to go one step further from the generic model we incorporated previously published energy matrices for RNAP and CI (Kinney *et al.* 2010 for RNAP; Sarai and Takeda, 1989 for CI) into the thermodynamic model. The only other aspect of the experimental system that we explicitly modeled were the cellular concentrations of RNAP and CI, which we estimated so as to maximize the fit between experimental and model estimates of epistasis. We did not explicitly model any other aspects of the system, namely the dimerization of CI molecules and the cooperative binding between CI dimers.

We quantitatively compare this model to our validation dataset of 30 mutants, which we first used to demonstrate that the sign of individual mutation effects alone is sufficient to predict the sign of epistasis (note that for the qualitative validation of the predictions arising from the generic model, namely that sign of epistasis can be predicted from sign of individual mutations and repressor concentration, we used the published energy matrices for RNAP and CI to determine the single and double mutant effects in order to validate our model. For RNAP, this is in contrast to what we presented in the first submission, where we obtained single mutation effects from our experimental measurements of the random mutant library). Using a linear regression, we evaluate how well the thermodynamic model with energy matrices describes the magnitude of epistasis, and find a significant fit both in the absence and in the presence of CI.

*3) In the box (which may be instead re-purposed as a Figure that could go with the new theory section), Panel b is rather perplexing (and needs to be explained better). The authors plot Gene expression (Shouldn't this be fold-change in expression?) as a function of Ep (which, if interpreted correctly, is the free energy of binding of the polymerase). The plot is unusual because the free energy goes from +0.05 to +10 (in units of kT?). A positive free energy represents the case of repulsion between polymerase and the promoter. While technically the plot is correct, in that for a repulsive free energy, the higher the energy the lower expression would be, that is never the case and all known binding free energies between RNAP and promoters are of course negative and thus favorable (see Bintu et al., Dodd et al., and a long etc.). This point is particularly unclear in its relation with the rest of the paper, so it should be clarified.*

The reviewers are right to observe the lack of clarity in the panel b of the box (now Figure 5), which stems from a silly mistake we made in the figure. Namely, we wrongly stated that the x-axis marked the binding energy. Instead, the plot was meant to display gene expression as a function of the change in binding energy, relative to the reference sequence with strongest possible binding. In this formulation, the sequence with strongest possible binding would have the value of 0. We introduced these changes in the figure and the figure legend. Furthermore, panel b) is meant as an illustration of how the general shape of the gene expression predicted by the thermodynamical framework leads to the patterns of epistasis that we find (in the absence of repressor). Hence, the units are not important but they are indeed units of kT.

*4) A second panel to Figure 2 should be presented containing the +CI data.*

We agree with the reviewers that the information about epistasis in the presence of CI was too obscurely concealed within Figure 4, and have added it to Figure 3 as a second panel.

*5) The authors should explain in Figure 4 why they synthesized 30 new variants, 17 with statistically significant epistasis, to test the model? Why not use all 141 mutants, and use the single mutants to predict the doubles? The authors should explain their logic for doing this, particularly in relation to how quantitatively calibrated their model is.*

Following the suggestions of the reviewers, we have now expanded the section ‘Independent validation of the generic model predictions’, the legend to Figure 6 (old Figure 4), and the accompanying Materials and methods section, in order to better explain why we use a separate dataset to validate the model. To summarize, the main finding of our paper is that for *cis*-regulatory elements, the biophysical properties of protein-DNA binding provide a null-model for epistasis, which results in the dependency of epistasis on TF concentrations. The generic model we use, which mathematically demonstrates why this is true for CREs in the absence of secondary effects of mutations, was inspired by experimental observations of 141 mutants. Most of these mutants had mutations in the regions of the CRE that were poorly described by the previously published energy matrices which we use in the model. In order to conduct an independent test of generic model predictions, we needed to introduce specific energy matrices for RNAP and CI, and hence to construct a new dataset where all mutations would be in the regions of the CRE that are well characterized by available energy matrices. As we have not picked the original 141 mutants in this manner, but rather randomly, incorporating them in the model would not provide an independent test of all the assumptions of the model.

*6) The concept of sign epistasis should be better explained and motivated in the introduction. For example, when it is first introduced (in the Introduction section), it is not defined. For many readers, this will make it very difficult to understand the paper. The importance of sign epistasis is detailed in the second paragraph of the Discussion. As this is important to motivate the work, the authors should move it to the introduction.*

In order to better explain sign epistasis and to motivate its significance, we introduced a new Figure 1, which provides a definition of the main types of epistasis (magnitude, sign, and reciprocal sign epistasis). In the Box we explain why we utilize the multiplicative instead of the additive model of epistasis to analyze this dataset and the model. In addition, we elaborated in the introduction on why sign epistasis matters for understanding evolution.